# Feasibility of Low-Cost LiDAR Scanner Implementation in Forest Sampling Techniques

**Michał Brach [1,\*], Wiktor Tracz [1], Grzegorz Krok [2] and Jakub Gąsior [1]**

[1] Department of Geomatics and Land Management, Institute of Forest Sciences,
Warsaw University of Life Sciences, 159 Nowoursynowska St., 02-776 Warsaw, Poland

[2] Department of Geomatics, Forest Research Institute, 3 Braci Leśnej Street, 05-090 Sękocin Stary, Poland

\* Correspondence: michal_brach@sggw.edu.pl

**Abstract:** Despite the growing impact of remote sensing technology in forest inventories globally, there is a continuous need for ground measurements on sample plots. Even though the newest volume assessment methodology requires fewer sample plots, the accuracy of ground-recorded data influences the final accuracy of forest stand modeling. Therefore, effective and economically justified tools are in the continuous interest of foresters. In the presented research, a consumer-grade light detection and ranging (LiDAR) sensor mounted on iPad was used for forest inventory sample plot data collection—including tree location and diameter breast height. In contrast to other similar research, feasibility and user-friendliness were also documented and emphasized. The study was conducted in 63 real sample plots used for the inventory of Polish forests. In total, 776 trees were scanned in 3 types of forest stands: pine, birch, and oak. The root mean square error was 0.28 m for tree locations and 0.06 m for diameter breast height. Various additional analyses were performed to describe the usage of an iPad in tree inventories. It was contended that low-cost LiDAR scanners might be successfully used in real forest conditions and can be considered a reliable and easy-to-implement tool in forest inventory measurements.

**Keywords:** iPad; LiDAR; forest; inventory; accuracy

## 1. Introduction

Reliable information on forest resources is essential to assess forest development. This information is traditionally collected during the National Forest Inventory (NFI) through ground surveys on a network of permanent fixed-area concentric sample plots. Systematic sampling is most often implemented as opposed to random sampling due to its advantages in practicality, data quality, and cost implications [1]. The benefits of systematic sampling for forest inventories mean that this is now standard international practice [2]. In many European counties, a design-based (probability-based) approach is used during forest inventories [3,4], and several important forest characteristics are estimated.

Several important forest characteristics and tree attributes measured on NFI sample plots are required for the mapping of forests and description of ecosystem services, such as tree species, tree height, diameter at breast height (DBH), tree location, and tree count. Different allometric models allow us to determine tree attributes that are not directly measurable but can be derived from other basic measurements. The inventory methodology was also designed to determine several important environmental parameters, which are land use/cover area, growing stock volume (GSV), and biomass. For example, GSV is a traditional indicator of wood resources, carbon stock, management efficiency, and sustainability in the forest sector [5–7]. GSV-related indices (biomass or carbon) must also be reported according to international agreements.

During forest inventories, accurate tree position measurements are necessary to determine whether a tree is within a particular plot and whether it is included in the sampling

scheme [8]. The number of trees on a plot is directly related to stand density and determines the spacing between trees. Tree spacing, in turn, indirectly indicates the availability of resources and aggregates the impact of various environmental factors on tree growth and physiology. This relationship between tree spacing and resource access translates to larger trees requiring greater amounts of resources for vigorous growth. The inverse relationship between stand density and tree size implies that decreasing stand density increases the mass of individual trees [9]. With tree size being a fundamental driver of timber value, information on stand density (spacing) and the number of trees allows forest managers to control the value of timber. Spacing between trees and the number of trees can also be used to estimate standing timber volume and value, help prescribe silvicultural treatments, and provide simple concepts for tracking forest growth dynamics.

The quality and quantity of the field inventory measurements are also important factors influencing forest inventory accuracy in large forest areas. Inventory results can often be improved by increasing the number and size of sample plots. However, practical considerations often limit the number of sample plots due to the cost of implementation or the complexity of the forest topography, especially in natural forests [10]. Due to sample size constraints, the inventory approach used in NFI may be sufficient to describe the entire population for large areas (countries, regions, provinces), but this approach may lack the precision to sufficiently describe forest/stand characteristics for small areas, especially areas with low sample coverage [11].

In general, traditional fieldwork in forest inventories is labor-intensive, time-consuming, and prone to different measurement errors [12]. Throughout the years, more automated and cost-effective techniques have been developed to address the shortcomings of traditional sampling methods. Due to the application of LiDAR (Light Detection and Ranging) instruments to a wide practice, forest metrics can be accurately and automatically derived from point clouds achieved with several LiDAR techniques, including TLS, ALS, MLS, and PLS.

Airborne Laser Scanning (ALS) has been used widely in forestry [13,14]. However, ALS may not be suitable to measure individual tree parameters due to lower point cloud densities (comparably to TLS or PLS) and crown obstructions of objects located under tree canopy [15]. Besides the possibility of precise measurement of basic tree attributes, such as DBH, tree position, and biomass [16–18], Terrestrial Laser Scanning (TLS) also has some disadvantages. Due to occlusion effects, 66% of all trees in the sample plot are not scanned from the plot center [19], and several scanning positions are necessary in order to measure all trees in a plot [20–23]. Mobile Laser Scanning (MLS) and Personal Laser Scanning (PLS), whose main advantages are rapid data collection from many positions, were introduced, and their applications have been studied in forestry [23,24]. MLS and PLS also have disadvantages, which include the high price of scanners, large device sizes, relatively labor-intensive data collection, complex data processing, and limited mobility of the MLS platform in a forest environment.

Another technique worth considering is the use of mobile devices such as tablets or smartphones to acquire a point cloud from the ground level. The use of such devices to generate point clouds in forest environments has been tested in previous years [25,26]. At that time, the technology developed in the Google Tango project was used, based on the use of digital images with additional information from a so-called depth map (RGB-D, or RGB + depth) generated from an additional camera capturing images in the infrared band.

In the work of Tomaštík et al. [25], data were acquired using a Lenovo Phab 2 Pro smartphone for 3 sample plots with a radius of 12.62 m. The average plot measurement time was 15–20 min. The study detected all the trees in the point cloud, obtaining an average error for the breast height measurement between 1.83 and 1.91 cm. The authors of the study also pointed out that the data acquired with this technology do not represent the upper parts of the stems. It is also worth noting that during the data acquisition step, two out of six attempts failed due to insufficient RAM on the device, which is needed for temporary data storage. This problem would likely be solved today by the technical

performance of more modern mobile devices. The authors also remarked on the relatively high energy consumption of the device during data acquisition, which was minimized by using external power bank batteries.

The study by Hyyppä et al. [26] dealt with information acquisition at the individual tree level. In the study, a point cloud was acquired using a smartphone for 80 trees, and each tree was scanned 3 times to check the repeatability of the estimated results. The average error in the estimation of breast height from the point cloud was 0.73 cm and compared to an analog reference tape measure. The authors tested the repeatability of the results and showed significant differences between measurements of the same tree. Differences between 0.32 and 1.47 cm were obtained for repeated measurements made on the same tree. The study also tested data acquisition capabilities for groups of trees. In the experiment, the researchers scanned two neighboring trees during a single pass. Based on the results obtained in this experiment, the authors did not recommend this technology for data acquisition on groups of trees. When analyzing the data for pairs of trees, they obtained an error in breast height estimation between 0.44 and 6.06 cm.

The results obtained in the referenced work are promising, especially considering the size and price of mobile devices. Unfortunately, the Google Tango project ceased development in 2018, resulting in no new devices supporting the technology. However, a new Google ARCore project is currently being developed to continue the achievements of the earlier project.

Recent studies on the possible usage of iPad Pro and iPhone Pro models give promising results which have the potential to change the approach to forest inventories [8,27–29]. However, most of this research focuses on the accuracy of tree diameter and overall tree detection. There is still a lack of research about absolute tree positioning using low-cost LiDAR scanners. Determining accurate tree position is a crucial factor that helps to match the ground data with ALS data, which in practice influences the correct biomass modeling [30].

The main goal of this research was to answer the question of whether low-cost LiDAR technology can be a reliable source to accurately measure tree position on inventory sample plots and conduct diameter breast height (DBH) measurements. The conducted fieldwork also allows us to analyze the user experience and weigh the pros and cons of capturing data via iPad. This work was validated due to the large volume of existing sample plot data in the National Tree Inventory in Poland used as reference data.

## 2. Materials and Methods

The research was performed on 63 selected sample plots located in the Jabłonna and Drewnica Forest Districts. These forest districts are located in the central part of the Mazowieckie Voivodeship and cover a fragment of the northern part of the Warsaw metropolitan area. The areas of both forest districts are part of the Promotional Forest Complex "Lasy Warszawskie", which is the buffer zone for Warsaw's intensive urban extent (Figure 1).

This area is mostly covered with coniferous forests, which are not very diverse in terms of species composition. Both forest districts in this study can be characterized as containing a vastly greater coniferous component than other forest types and a minority of other mesic habitats. Reference information on the research areas was obtained on the basis of data provided by the Bureau for Forest Management and Geodesy. The data come from measurement works that were carried out in the years 2017–2021 as part of the National Forest Inventory. Each of the sampling points was a fixed-radius sample plot with a radius of 11.28 m and an area of approximately 400 m$^2$. For this study, trees located on research plots with recorded DBH measurements at a height of 1.3 m on the stem, and exceeding 70 mm in diameter, were used in the analysis. The study plots provided representative diversity in terms of forest habitat type, stand structure, landform, species composition, number of trees, and tree age. Scots pine (*Pinus sylvestris* L.) was the dominant species on over 70% of all study plots. Silver birch (*Betula pendula Roth*) was the dominant species in

10% of the area, while oak (*Quercus spp.* L.) had a share of 5%. Other species prevailing in the stands are common elm (*Ulmus minor Mill.*), black alder (*Alnus glutinosa* (L.) *Gaertn.*), aspen (*Populus tremula* L.), and American ash (*Fraxinus americana* L.). Depending on the location in the field and the type of forest habitat, the study plots were generally diversified regarding the number of species in the sampling area. Sampling area stand ages were strongly differentiated, from 15 to even 134 years, with the vast majority of stands in intermediate age, i.e., between 60 and 75 years of age. The number of trees on each plot ranged from 7 to 37 (Table 1).

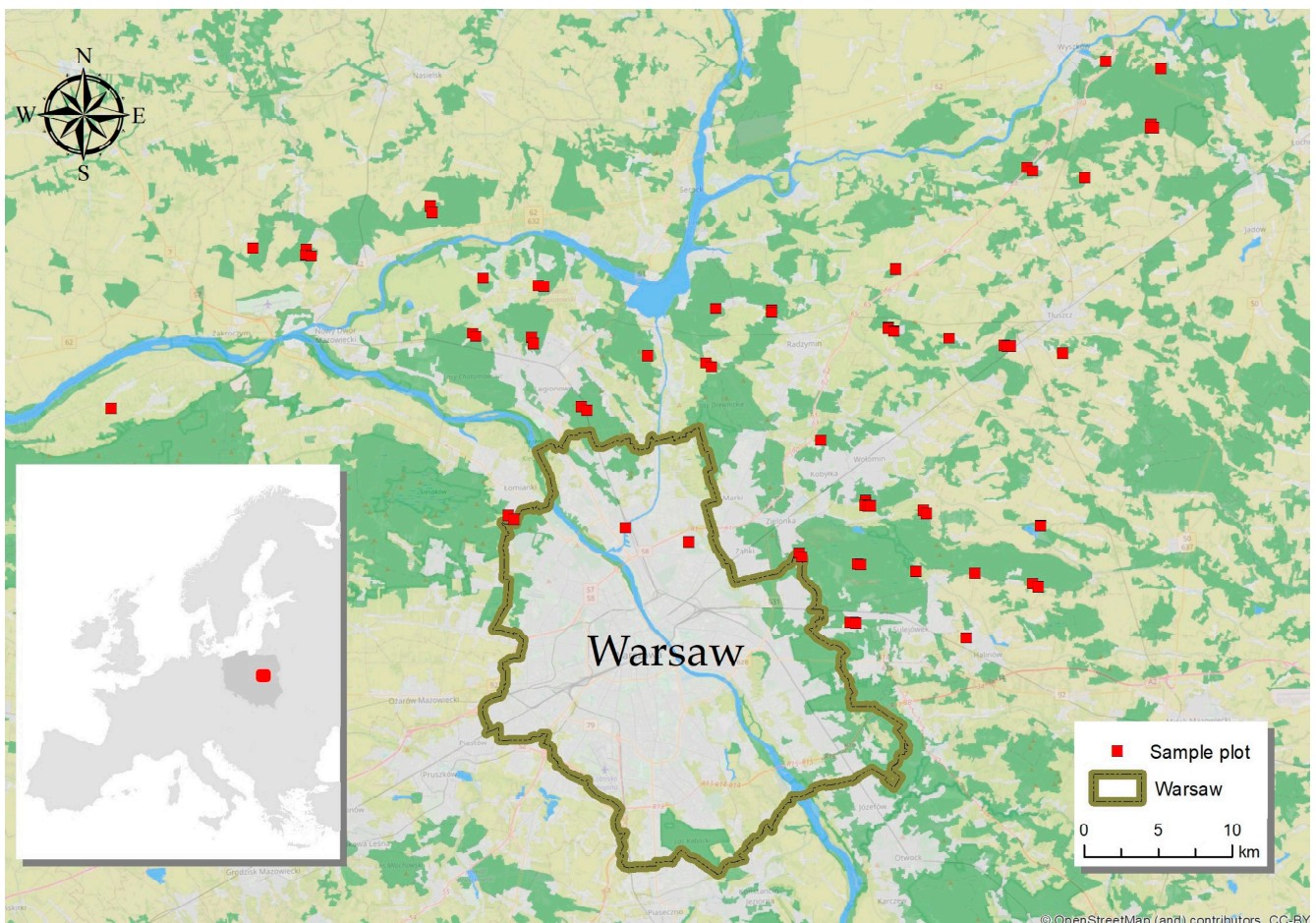

**Figure 1.** Location of sample plots around the capital of Poland—Warsaw.

**Table 1.** Detailed characteristics of the main types of stands occurring on the study plots.

| Stand Type | Pine | Birch | Oak | All |
|---|---|---|---|---|
| Number of plots | 49 | 9 | 5 | 63 |
| Average age | 65 | 58 | 57 | 63 |
| Average height (m) | 18.63 | 15.14 | 19.03 | 18.16 |
| Average number of trees | 14 | 10 | 13 | 13 |
| Average DBH (cm) | 25.9 | 21.48 | 27.27 | 25.38 |

Most of the study plots were located in areas with flat topography, while plots located in areas with dune or hilly topography were a minority. The diversified structure of research areas ensures the suitability of this research and enables the presentation of real possibilities of using terrestrial smartphone-based laser scanning technology in forestry works in relation to field conditions typical for most of Poland.

According to the methodology of the National Forest Inventory, the middle of the sample plot was measured by the geodetic class GNSS receiver. The accepted measured

mode was FIX with a connection to the network of reference stations ASG-EUPOS and real-time kinematic data capture. Taking into account the technical specification of the ASG-EUPOS and GNSS receiver used, the horizontal accuracy should not exceed $\pm 0.03$ m [31]. The middle of the sample plot was the base for the classic polar surveying method used for determining the tree locations. The azimuth was measured by the Tracon Ushikata surveying compass (Survey Supply, Inc., Milford, CT, USA) with a 1-degree accuracy. The distance was measured by Vertex IV (Haglöf Sweden AB Långsele, Sweden) with an accuracy of around 0.01 m. The DBH was checked in two directions (parallel and perpendicular to the middle of the sample plot) by the Haglöf Digitech BT caliper with a resolution o 0.001 m. With respect to the sample plot radius and the surveying compass accuracy, it can be expected that the maximum error for the trees located on the border of the sample plot should not exceed 0.20 m [32]. The complete documentation of the tree locations was stored in the Polish coordinate system 1992 (EPSG 2180) and served as the reference for further calculations and analysis.

The field scanning and data collection via iPad was conducted over two months (October and November 2022). Being the autumn season in Poland, there were no leaves on the first floor of the sample plots. The weather was generally cloudy, with temperatures of around 6 °C and minimal/no wind conditions. The iPad Pro generation 3 (Apple, Cupertino, CA, USA) with iOS 15.6 operating system and 256 GB internal memory was used as the main tool for scanning the sample plots. The device is equipped with a consumer-grade LiDARscanner, which, combined with a high resolution onboard 12 MP RGB camera, allows us to capture three-dimensional objects within the 5 m range. A low-cost LiDAR scanning unit consists of two integrated elements: vertical-cavity surface-emitting lasers (VCSELs) produced by Lumentum (Lumentum Operations LLC, San Jose, CA, USA) and near-infrared CMOS receptor produced by Sony (Sony Group Co., Tokyo, Japan). This pairing of a low-cost scanner with a high-resolution onboard RGB camera (12 MP) allows the capture of 3-dimensional objects within a 5 m range through the use of the direct time of flight (dToF) technology [33]. Further application of stereo matching methods [34], positioning data from an integrated GNSS (Global Navigation Satellite Systems) receiver, and an internal measurement unit (IMU) [27] to the scanning data allows for the creation of a fully colored 3D point cloud. There are many free applications that can be successfully used for the 3D inventory. Based on a literature review [27,28,35–37] and field experience, it was decided to use 3D Scanner App version 2.0.8 (Laan Labs, New York, NY, USA). This software is characterized by a very simple user interface and a huge number of file formats for data export and was confirmed by Gollob et al. [8] as the best solution for iPad. Considering the relatively short days during the autumn season in Poland and the location of sample plots, it was possible to capture a maximum of 9 scans daily. Outdoor temperature conditions were favorable, and there were no problems with battery life. Additional charges were not required. The total data storage requirements for 63 sample plots was approximately 180 GB, which did not influence the iPad function or software efficiency.

Before the scanning procedure, one reference sign was located in the middle of the sample plot, and the second one was at a distance of five meters to the north or to the south, depending on the tree's location. Additionally, four measuring tapes were used to mark the 11.28 m plot radius. The modes "advanced" and "GPS tag" were turned on in the 3D Scanner App, and the LiDAR scanned around 3 m to avoid the accidental reading of other trees. Every scan was started by the data collector from a reference sign, and afterward, the data collector continuously walked around all trees with a DBH larger than 0.07 m. If the distance between trees was further than the scanner range, the ground was scanned until the next tree appeared in the range of the 3D Scanner App. This methodology was confirmed by Mokroš et al. [29] and allowed the creation of a continuous model of the sample plot. Every tree was rounded and scanned, starting from the bottom of the tree trunk and ending at a height a little higher than 1.3 m to correctly register diameter at breast height. The single continuous walk across sample plots for data collection lasted around 10 min, depending on the forest conditions. There were no situations that demanded

scanning repetition because of RAM (Random Access Memory) or CPU (Central Processor Unit) usage. There was also no single crash of the 3D Scanner App (Figure 2). At the end of the continuous scan, the reference sign located in the middle of the sample plot was scanned again. The post-processing of data by the 3D Scanner App took around 3 min and could be realized in the background of the operating system. iTunes software was used to transfer high-density LAS Geo-Referenced, and image files to the desktop computer. The default coordinate system for the scans is WGS84, so in order to match the results with reference, the files were transformed to Polish coordinate system 1992 (EPSG code 2180) by the lidR–R package for Airborne LiDAR Data Manipulation and Visualization for Forestry Applications [38]. The complete vector walk lines were extracted from the .json files assigned to images by the dedicated Python script.

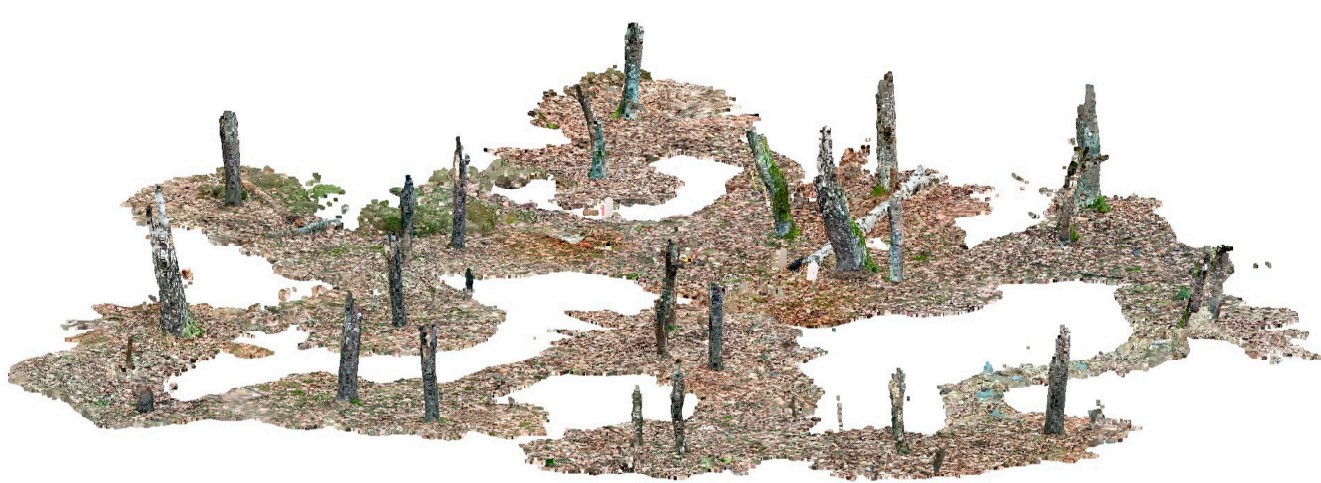

**Figure 2.** Visualization of point cloud LAS file captured by 3D Scanner App for one of the sample plots. Data are presented in the CloudCompare software.

The process of segmenting the stems from the point cloud was performed automatically in the R development environment using the lidR library [38] and external tools such as CloudCompare 2.12.1 software (The CloudCompare open source project, Grenoble, France) [39]. In the first step, the input point cloud was segmented into two clusters—the ground and everything above the ground (stems). Then, the point cloud on the cluster representing the trees was divided into individual stems using the connected components algorithm. The final step was to use the RANSAC circle fitting algorithm to slice the stems 1.25–1.35 m above the ground and measure DBH. In addition, the Circumferential Completeness Index (CCI) was used in order to assess if the whole trunk of the tree was scanned by iPad [40]. Using the information on the axis coordinates and diameter obtained from the application of circle fitting with the RANSAC algorithm, the point cloud slice was divided into 16 angular zones (each zone occupied a slice of 22.5°) that divided the stem into equal zones along a circle. In each zone, points from the point cloud slice were assessed to see if they were within the zone. If there were points in a zone, the zone was assigned a value of 1; if not, a value of 0. In the end, the values of all zones were added together, and the ratio of filled zones to unfilled zones was determined. If all zones were filled, the CCI value was 1. The cutoff values for the zones were 0.7 times and 1.3 times the stem diameter estimated by the RANSAC algorithm. The CCI calculations were performed in the R environment using a custom implementation of the algorithm. This factor was used in parallel for validation of the diameter breast at height measurements. The reference data from the sample plot were compared to the matched trees from the scans, including the CCI and the year of sample plot reference measurements.

The validation of datasets is based on the root mean square error (RMSE), which describes the real distance between the reference trees and trees extracted from the scans [41]. Additionally, the mean absolute error (MAE) was calculated in order to express the bias

value independently for X and Y coordinates. For the precision assessment of tree positioning, the standard errors (SE) were used, which remove the bias error for the datasets [42]. The normality of results was checked by the Shapiro–Wilk test [43]. The significance of the results according to the sample plots variables was checked by Kruskal–Wallis test [44]. In the case of significant differences (*p*-value smaller than 0.05), the Dunn test was performed for further analysis [45]. All statistics were performed by the R package [46].

### 3. Results

The GNSS receiver in iPad assigns coordinates to the scans with a mean RMSE error equal to 2.71 m. The center point of the sample plots recorded in the scans was mostly located in the southwest direction with a mean azimuth of 218 degrees starting from the reference center of the sample plot (Figure 3a). The location of trees extracted from the scans was matched with the reference data, and only this pair of trees was passed up for accuracy analysis. The difference in the tree diameter between scanned and referenced trees results from differences in the age of the trees (Figure 3b).

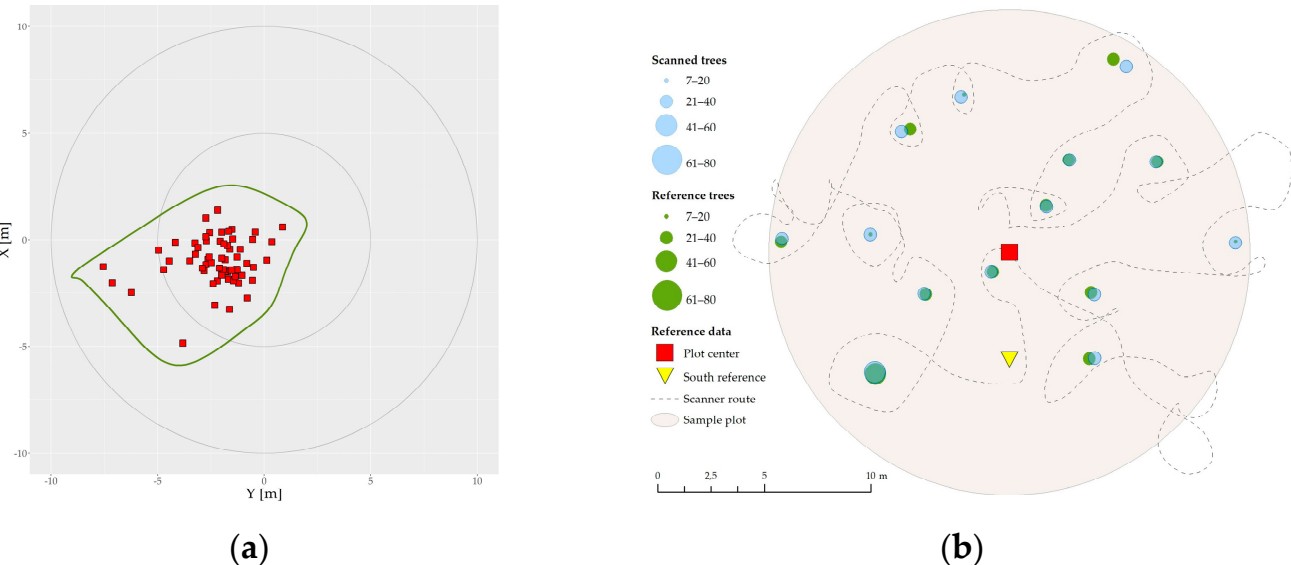

**(a)**  **(b)**

**Figure 3.** Sample plot schemas: (**a**) diagram of the sample plots center extracted from the scans (red square) according to the real sample plot location; (**b**) the vector visualization of one of the sample plot data sets with scanned and referenced trees represented by the circle with diameter breast height expressed in centimeters.

Considering the linear reference delineated between the center of the sample plot and the north/south point, all scans were rotated with a mean of 7.68 degrees to the left. In total, 776 trees were extracted from the scans captured by iPad Pro. The most common main tree species was pine, with 626 scanned trees, and 88 for birch and 62 for oak, respectively. Over 15 km was covered through 63 sample plots, which represent very different tree conditions. The route length was significantly dependent on the number of trees (*p*-value = $2.59 \times 10^{-5}$, r Pearson = 0.50) but was not significantly connected with the tree's age (*p*-value = 0.56, r Pearson = 0.07). The mean distance of the route was 244 m, and 50% of all routes ranged between 200 and 300 m (Figure 4).

The root mean square error was 0.28 m for all trees; however, the biggest difference between the first and third quartiles was observed for oak and was equal to 0.32 m. These same tendencies were found for the mean absolute error and standard error, where the oak trees' location of the scans was characterized by the biggest dispersion, which was 0.21 m for the MAEx, 0.23 m for the MAEy, and 0.10 m for both SEx and SEy errors, respectively (Figure 5, Table 2).

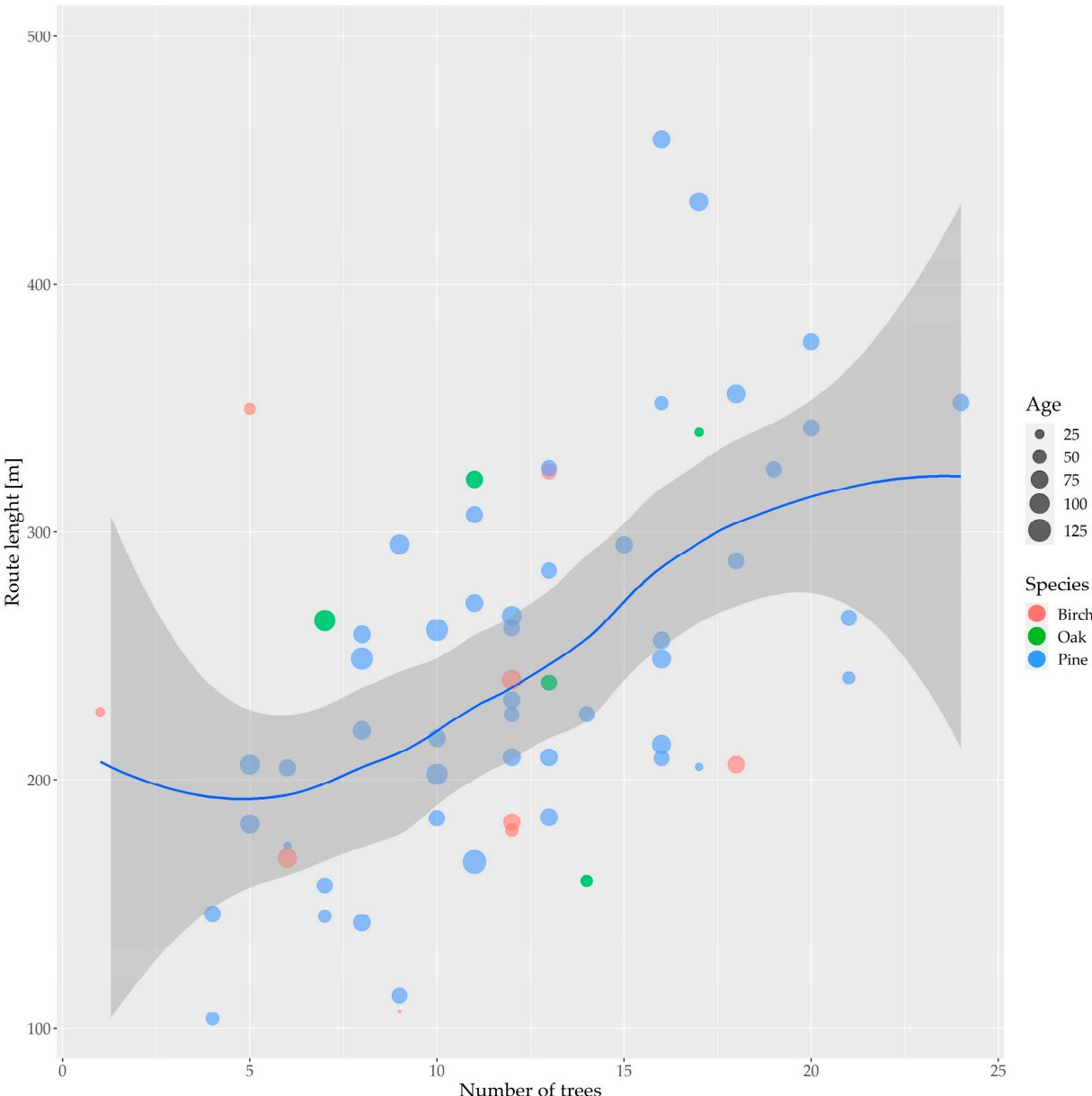

**Figure 4.** Dependence of the number of trees according to the route length with the scanner. The dot size presents the main tree species' age, and the color indicates the species.

**Table 2.** Comparison of errors taking into consideration the main tree species.

| Variable | Pine | Birch | Oak | All |
|---|---|---|---|---|
| Number of trees | 626 | 88 | 62 | 776 |
| RMSE (m) | 0.28 | 0.25 | 0.39 | 0.28 |
| Maximum RMSE (m) | 0.81 | 0.63 | 0.81 | 0.81 |
| MAEx (m) | 0.15 | 0.14 | 0.21 | 0.15 |
| MAEy (m) | 0.14 | 0.12 | 0.21 | 0.15 |
| SEx (m) | 0.20 | 0.19 | 0.25 | 0.20 |
| SEy (m) | 0.19 | 0.16 | 0.28 | 0.20 |

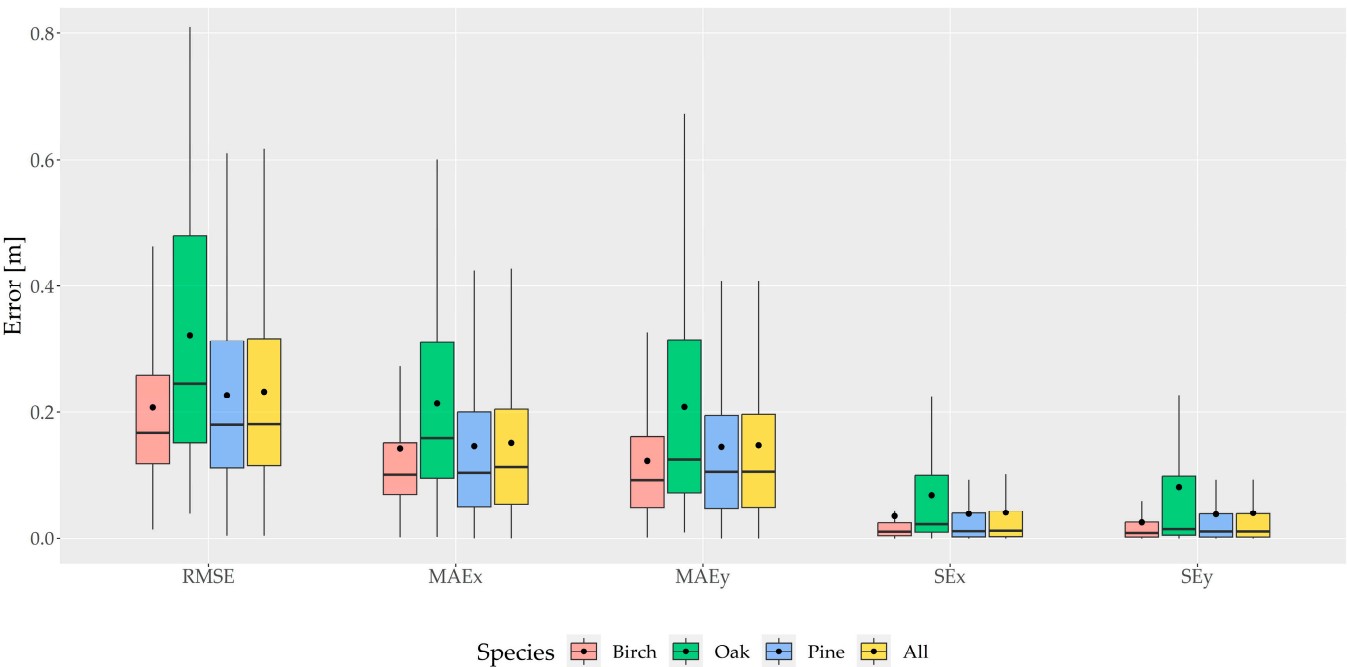

**Figure 5.** The boxplot represents the errors, including the type of error and main tree species. The black dot symbolizes mean value.

Less than 240 (31%) trees had an RMSE bigger than the mean, but only 69 (9%) trees scanned had an RMSE bigger than 0.5 m. The Shapiro–Wilk test confirmed that none of the errors had a normal distribution (Table 3). In almost all cases, the MAE error for the X and Y coordinates was equal to half of the RMSE error. Taking into account the standard error definition, which removes the bias, the final precision of tree positioning of trees on the scans was much smaller and equal to 0.20 m for both X and Y coordinates.

**Table 3.** Results of Shapiro–Wilk test (expressed in *p*-value) when checking the normal distribution of errors while taking into consideration the main tree species.

| Variable | Pine | Birch | Oak | All |
|---|---|---|---|---|
| RMSE | $2.2 \times 10^{-16}$ | $4.913 \times 10^{-7}$ | $9.549 \times 10^{-5}$ | $2.2 \times 10^{-16}$ |
| MAEx | $2.2 \times 10^{-16}$ | $4.026 \times 10^{-10}$ | 0.001899 | $2.2 \times 10^{-16}$ |
| MAEy | $2.2 \times 10^{-16}$ | $1.711 \times 10^{-7}$ | $1.486 \times 10^{-7}$ | $2.2 \times 10^{-16}$ |
| SEx | $2.2 \times 10^{-16}$ | $3.149 \times 10^{-15}$ | $9.504 \times 10^{-9}$ | $2.2 \times 10^{-16}$ |
| SEy | $2.2 \times 10^{-16}$ | $1.108 \times 10^{-14}$ | $1.962 \times 10^{-10}$ | $2.2 \times 10^{-16}$ |

The Kruskal–Wallis test proved the significant dependency between all errors and main tree species. Hence, the Dunn test with Bonferroni correction was performed for all combinations of errors and main tree species. A significant difference was found only between oak and other species (birch, pine), where the *p*-value was smaller than 0.05.

For every scanned tree, the azimuth to the matched reference tree was also investigated. The Shapiro–Wilk test allows rejecting the null hypothesis about azimuth normality (*p*-value = $6.483 \times 10^{-16}$). There was also no correlation (r Spearman = 0.041) found between the azimuth value and the RMSE for all tree species. A weak correlation was found for oak (r Spearman = 0.322), but it should be mentioned that oak was represented by a smaller number of analyzed trees. The Kruskal–Wallis test confirmed no significant differences for the azimuth of scanned trees to matched reference trees when considering the species samples (Figure 6).

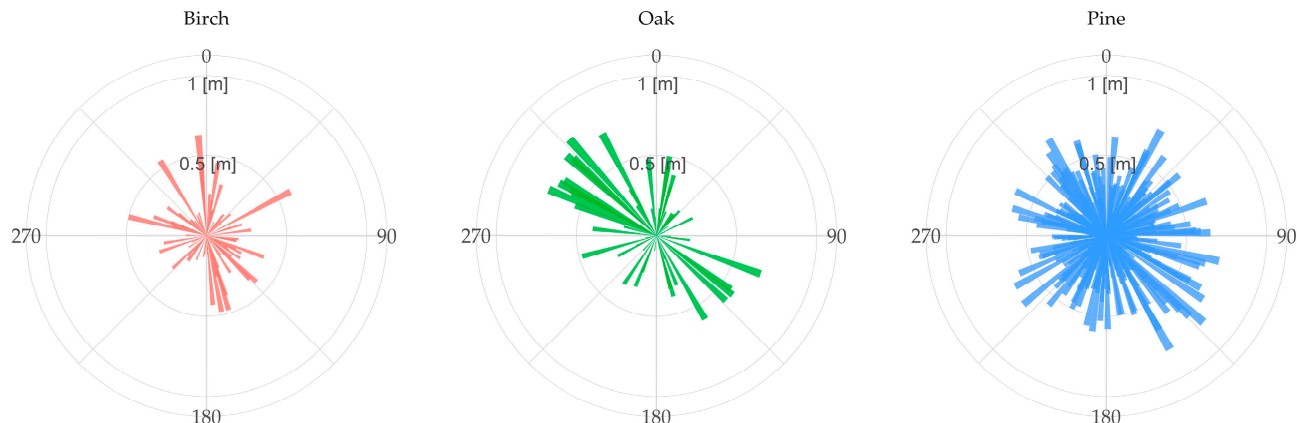

**Figure 6.** The RMSE value (expressed in meters) and the azimuth (expressed in degrees) for every scanned tree to the matched reference trees while taking into consideration tree species.

The reference DBH was measured in two directions; hence, only CCI bigger than 0.75 was under further analysis, which in practice guarantees that 75% of the tree trunk was covered by the point cloud. More than 80% of the analyzed dataset reached 75% point cloud coverage for the diameter breast height, and 53% for the CCI equaled 1, respectively. The RMSE for the bigger sample of trees characterized by the 0.75 CCI was 0.06 m, and 0.05 for the CCI equaled 1. The coefficient of determination reached 0.85 for all trees with correctly extracted diameter and 0.9 for the tress fully rounded by the point cloud at diameter breast height (Figure 7).

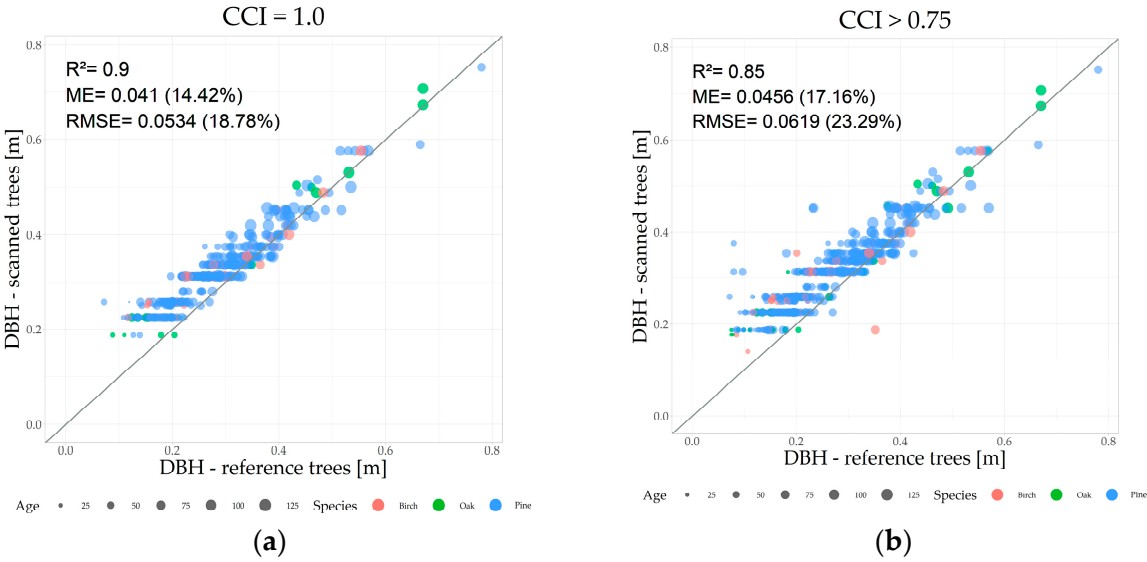

**Figure 7.** Correlation between DBH extracted from the iPad scans and the reference DBH, including the main tree species and trees age: (**a**) Circumferential Completeness Index equal to 1; (**b**) Circumferential Completeness Index equal to 0.75.

## 4. Discussion

The low-cost LiDAR scanner implemented in iPad Pro has been researched in a few articles [8,27–29,35–37]; however, none of the cited articles contained sample datasets of this size and scope while also representing various stand types. One of the most important reasons to act on this research was a productive approach to the low-cost scanner. As was mentioned in the Methodology, the current methods of tree inventory on the sample plot level are not sophisticated. The only crucial modification in place is the implementation of the GNSS receiver for the determination of the X and Y coordinates of the sample plot

center. The GNSS mostly operates in the RTK mode; however, with a lack of GSM coverage and dense tree coverage, the availability of centimeter accuracy is still low [47]. This has been a frequently mentioned result, even in the newest forest research [48]. With this result, the final positioning accuracy is still not reliable. This is exactly why the ground inventory should be conducted in the most accurate way possible. In practice, the tree locations are based on the azimuth determined by the surveying compass, so we cannot expect centimeter accuracy. Additionally, the sample plot documentation is based on the tree location schema and one image taken from a random location with the compass located at the plot center. In such circumstances, there is no doubt that three-dimensional documentation can improve not only the location of the tree but can be a perfect supplement to detailed sample plot documentation as well.

The 63-sample plot investigated in the presented research was visited for the first time with a low-cost scanner. At the end of the field measurements campaign, it was concluded that the delivered documentation was not enough to find the real sample plot locations. The 3D model displayed on the tablet with connection to GNSS positioning helped tremendously to find the sample plot center point and point previously mapped trees. This idea is already implemented by the viDoc RTK rover (Pix4D S.A., Prilly, Switzerland), which is a specially created device for positioning augmentation of iPad/iPhone low-cost scanners. Implementation of a surveying-grade GNSS receiver, which works in RTK mode, allows precise point cloud positioning in the required coordinate system. However, keeping in mind the strong multipath effect under the forest canopies [49,50], which is hard to mitigate in an obstacle-rich environment [51], the progress in positioning accuracy will still be low. Considering the root mean square error extracted from the mobile class GNSS receiver mounted in an iPhone, the range of 7–13 m can be expected depending on the condition of data capture [52,53]. The 10 min observation by a smartphone in full canopy cover results in 3.68 m horizontal accuracy [54]. Compared to these results, the performance of the iPad was better than expected (2.71 m RMSE). It should also be pointed out that the iPad was in continuous movement, which can be an additional factor of positioning degradation [55]. The archived results are convergent with Corradetti et al. [56], who mentioned that the accuracy of the iPad GNSS receiver was within a few meters thanks to the Apple Core Location framework. Most of the center plot coordinates were oriented in the southeast direction, so the character of this error is systematic (Figure 3a). It could be observed that the route length with the scanner was dependent on the number of trees within the sample plot. In practice, the most problematic obstacles during the scanning process were the presence of the bushes and dead fallen trees, which were omitted. Such conditions can potentially occur on every sample plot, which is why the significance of tree age for the route length is low (Figure 4). Additionally, the operator with the iPad was mostly focused on the screen, not on the route difficulty, and hence having a second person who can help to clear the way would be relevant. Nevertheless, before scanning, no special route was planned. Additionally, there is no need to worry about missing or repeated trees because all scanned data are visible live in the 3D Scanner App. One of the benefits of the 3D Scanner App is that it forces the operator to keep a stable walking pace. It is not possible to scan too quickly, as missed areas on the generated surface appear immediately. Hence, the scanning speed with the 3D Scanner App is very natural and is a product of the iPad's efficiency and safe operation in forest environments.

The overall RMSE for the trees extracted from the scans was 0.28 m. This error is related to the mean DBH of reference trees, which is 0.25 m. In practice, it means that the final position of trees on the scan is similar to the mean diameter of trees. Additionally, it is worth mentioning that the present methodology of tree location is a result of two errors: azimuth and distance measurements. Especially for long distances from the sample plot center, it can be expected that the final tree positions will be close to the iPad scans. The quality of low-cost scanners is slightly higher considering the standard error, which removes the systematic movement of trees. The improvement in this area is not high and is equal to 0.20 m for the X and Y coordinates, respectively. This is partially caused by the significant

diversity of movement direction according to the reference trees and confirmed by the normality results for all errors (Table 2). The statistical test confirmed that trees scanned by iPad could be located in every direction according to the reference (Figure 7). Nevertheless, this drawback has weak overall implications considering the RMSE value, which is very stable for all researched tree species. The regularity of data capture by low-cost scanner was confirmed by very similar mean values of all errors and was much better than the 3.1 m reported by Tatsumi et al. [57]. There was no correlation between the route length and RMSE; however, it can be expected that longer routes can lead to iPad overloading. However, when examining the mean size of sample plots optimal in forest inventory techniques [58], there is no concern that such problems can appear.

The presented research confirmed the high efficacy of DBH measurements using a low-cost scanner. The mean RMSE for DBH, including 75% coverage of the point cloud around the tree trunk, was 0.06 m. These results are comparable to other studies that reported RMSE error ranges from 3 to 8 cm [8,27,33,57]. The big advantage of the iPad in this area is the high percentage of point cloud coverage around scanned trees. More than 80% of scanned trees reached 75 CCI. Modeling site productivity [59] and reliable sample plot inventory needed for biomass assessments [60] mostly depend on accurate DBH measurements. When examining the achieved results, it can be assumed that a low-cost LiDAR scanner can be a reliable tool for diameter breast height measurements. Compared to traditional calipers, the presented technology is temporally much more efficient and can guarantee similar accuracy. One of the biggest advantages of the iPad compared to millimeter DBH accuracy achieved by TLS technology [61] is the tree horizon visibility. Especially in dense forest stands, the implementation of plot center-based TLS scanning is limited [22]. This problem is not present in the case of iPad usage, which allows for free movement and adjustment of sample plot size up to inventory requirements. The same low-cost LiDAR sensor is also mounted in iPhones, so in practice, the size of the equipment can be radically reduced without any impact on final accuracy [57]. In contrast to digital data capture methods, there are still many professionals who prefer classical measurement methods. A good example is Filed-Map (IFER–Monitoring and Mapping Solutions, Ltd., Jilove u Prahy, Czech Republic), which offers a professional hardware and software solution for tree inventory and visualization [62,63]. Both solutions have strong and weak points, so it can be expected that the solutions in the future will combine laser scanning with advanced software algorithms for biomass assessment.

In spite of the low-cost scanner's many benefits, several limitations were uncovered over the course of two months of measurement. First, if the scan route for the session is too long, the data collected will overload the operator's iPad. Second, to ensure accurate measurements, the operator, who must focus on the screen, will require an assistant to help navigate challenging understory conditions. This is especially true in dense forests or intense groundcover, where the operator spends more time navigating a safe route than focusing on the scanning objectives. An additional point of consideration is that it is simply exhausting to operate the iPad for more than 10–15 min in a state of intense concentration. These observations point to the conclusion that scanning by iPad is severely limited by vegetation in rich habitats. Furthermore, it was observed that bushes with heights close to DBH caused drastic signal interference when distributed around the trunks of trees.

A relatively more significant error value for the oak helped to point out the factors influencing the iPad's scanning accuracy, considering that the distance of the oaks from the plot center was noticeably longer compared to pine and birch. This is reflected in a bigger dispersion of the oak trees on the sample plot, which can increase the RMSE error (Figure 8a). Additionally, the age range for the oak trees is wider compared to other analyzed species, so it can be expected that trees with lower DBH values influence the final positioning accuracy (Figure 8b). Finally, the distance of analyzed trees along the scanning route was checked. This factor was extracted from the route line segmented by the measure value (M). The closest M value was assigned to every scanned tree within the sample plot (Figure 3b). It is clearly visible that most of the oaks were scanned after 100+ m of scanned

travel with an iPad. These values are relatively lower for pine and birch, which suggests that a longer scanning route increases the RMSE error (Figure 8c).

**Figure 8.** The boxplots for three groups of species containing: (**a**) the distance of the single tree from the sample plot center, which was also the start point of the scanning route; (**b**) the age of all trees within the species group; and (**c**) the distance on the scanning route for every analyzed tree, where a 0 value means the start of the scan.

In regard to data transfer and post-processing, several points of improvement should also be considered. The 3D Scanner App offers numerous formatting options but no ability to determine an extension for all files saved in the memory. What is worse is that iTunes, a proprietary apple application, is required for data transfer. Not only does this necessitate the use of a poorly optimized virtual iOS environment (in the case of Windows systems), but it also disallows the user a large degree of access to the iPad disc where the data are stored. Finally, from a processing point of view, the data required a high degree of manipulation in R software to be usable within the scope of this research. There is no ready-to-use application that can be applied to iPad point cloud processing. Nevertheless, the fact remains that the future of forest inventory is an open field for the application of low-cost scanners such as the iPad, and most, if not all, prior mentioned drawbacks of their use are easily mitigated.

## 5. Conclusions

The main goal of this research was to prove the feasibility of a low-cost LiDAR scanner as an effective tool for forest inventory. The implementation of iPad inventory data collection was conducted in managed forests that were under continuous silviculture treatment, so it can be expected that the final results will closely model the reality of everyday implementation in forest metric quantification. The RMSE equal to 0.28 m for tree position could be improved, but the open question is if there is a need to have better accuracy. Biomass will be, or already is, assessed by ALS data, so more importantly, it is necessary to correct DBH values and relative tree positioning in order to calibrate ground measurements with canopy height models. The low-cost LiDAR scanner successfully fills this area and also adds many supplemental advantages, such as three-dimensional sample plot documentation, fast data acquisition, high relative tree location, and high DBH assessment accuracy. There is no doubt that TLS performance is higher, but considering the cost of equipment, data processing needs, and pre-scan plot preparation, low-cost scanners can provide a capital-intensive alternative for many foresters to achieve the results they need to meet their management goals.

**Author Contributions:** Conceptualization, M.B., W.T. and G.K.; methodology, M.B. and G.K.; software, M.B., W.T., G.K. and J.G.; validation, M.B., G.K. and J.G.; formal analysis, M.B., W.T., G.K. and J.G.; resources, M.B., W.T. and J.G.; data curation, M.B.; writing—original draft preparation, M.B., W.T., G.K. and J.G.; writing—review and editing, M.B., W.T. and G.K.; visualization, M.B. and J.G. All authors have read and agreed to the published version of the manuscript.

**Funding:** This research received no external funding.

**Data Availability Statement:** The data for this research can be shared upon request to the author.

**Acknowledgments:** We would like to thank Dariusz Górski for his help in Python programming. We are also thankful for the perfect cooperation with Marcin Mionskowski from the Bureau for Forest Management and Geodesy in reference data understanding. Special thanks to Robert Magnuson for proofreading the final text version.

**Conflicts of Interest:** The authors declare no conflict of interest.

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
