# Peer review of "Feasibility of Low-Cost LiDAR Scanner Implementation in Forest Sampling Techniques"

_forests, doi:10.3390/f14040706_

Round 1

Reviewer 1 Report

Overall Summary Evaluation:

The manuscript is well written and the statistical knowledge is appropriately used. This study used a consumer-grade LiDAR sensor mounted on iPad to collect tree location and DBH data. Based on the scanning results, the RMSE was 0.28 m and 0.06 m for tree locations and DBH, respectively. The results and interpretations are well explained and the figures and tables are properly used.

However, a number of significant changes can be made in order to improve the general quality of the paper. Some comments or proposals are given below:

Comment 1: Line 83-84: “PLS also has its disadvantages which include high price of scanners, lower accuracy of output point clouds.” Current, the accuracy of output point clouds of some PLS is not very low.

Comment 2: Line137: “which is the buffer zone of the Warsaw which is the buffer zone”, please check the sentence.

Comment 3: Line150: “exceeding 70 mm in diameter, were used in the analysis”. Is the accuracy of this instrument feasible for trees with a DBH of less than 7 cm? Does the author plan to conduct corresponding verification in the future to improve the application scope of this instrument in forest inventory?

Comment 4: Based on the manuscript, the study area with flat topography and the stand structure is not very complex, while the sentence in the abstract section “trees were scanned in very different forest stand conditions”, you refer to the distribution of different tree species?

Comment 5: When scanning trees in the sample plot, does the author need to plan the scanning path in advance to ensure that all trees in the plots can be scanned? Since the instrument can only obtain three-dimensional object information within 5 meters, it is necessary to be as close to each tree as possible.

Comment 6: For the CCI, I suggest authors use some algorithms to improve the circumferential completeness when the scanning point cloud is incomplete, which will improve the estimation accuracy of DBH.

Comment 7: During the scanning process, will the speed of walking have a certain impact on the quality of scanned point cloud data?

Author Response

Dear Reviewer,
We were really happy for very positive feedback from you. This is important for me because the main reason for this article and generally for my research is to help practitioners foresters and researchers in their daily job. We would like also to declare that since the original version of the manuscript was improved what was possible by your and other reviews' comments. We also sent you the corrected version and the track version in order to see to help with the observation of changes. The text passed additional proofreading and we hope now it is much better. Once again we want to thank you very much – it was a pleasure for us.

all the best

Authors

Reviewer 2 Report

Feasibility of low-cost LiDAR scanner implementation in forest sampling techniques

This manuscript used a low-cost LiDAR scanner to accurately measure tree position on inventory sample plots. The results proved the feasibility of low-cost LiDAR scanner as an effective tool for the forest inventory. The manuscript is logically structured and reads well. The description of the objectives, methods and results is mostly clear and comprehensible. I think this manuscript has a minor revision. The detailed comments are given below:

Detailed comments:

1.       A few of typos, spelling mistakes etc. in the paper. So I recommended the author to carefully read through the draft to correct each typo.

2.       Please add a north arrow in Figure 1.

3.       Please improve the resolution of Figure 6.

4.       How do you calculate the metric of CCI? Please add more descriptions about the calculation of CCI?

5.       Please increase the text size in the Figure 7.

6.       Line 237: Please add the software company name after the “cloudcompare”.

7.       Line 244: “In addition, the Circumferential Completeness Index (CCI).”. This sentence seems incomplete.

8.     Please discuss the effects of the tree species on the results. From Figure 6 and Table 3, Oak had the lowest accuracy. Why?

Author Response

(The authors gave the same response as above.)
